# Intracellular Conformation of Amyotrophic Lateral Sclerosis-Causative TDP-43

**DOI:** 10.3390/ijms24065513

**Published:** 2023-03-14

**Authors:** Akira Kitamura, Sachiko Yuno, Rintaro Kawamura, Masataka Kinjo

**Affiliations:** 1Laboratory of Molecular Cell Dynamics, Faculty of Advanced Life Science, Hokkaido University, Sapporo 001-0021, Japan; 2PRIME, The Japan Agency for Medical Research and Development, Tokyo 100-0004, Japan; 3Laboratory of Molecular Cell Dynamics, Graduate School of Life Science, Hokkaido University, Sapporo 001-0021, Japan

**Keywords:** TDP-43, amyotrophic lateral sclerosis, protein structure, Förster resonance energy transfer, fluorescence correlation spectroscopy

## Abstract

Transactive response element DNA/RNA-binding protein 43 kDa (TDP-43) is the causative protein of amyotrophic lateral sclerosis (ALS); several ALS-associated mutants of TDP-43 have been identified. TDP-43 has several domains: an N-terminal domain, two RNA/DNA-recognition motifs, and a C-terminal intrinsically disordered region (IDR). Its structures have been partially determined, but the whole structure remains elusive. In this study, we investigate the possible end-to-end distance between the N- and C-termini of TDP-43, its alterations due to ALS-associated mutations in the IDR, and its apparent molecular shape in live cells using Förster resonance energy transfer (FRET) and fluorescence correlation spectroscopy (FCS). Furthermore, the interaction between ALS-associated TDP-43 and heteronuclear ribonucleoprotein A1 (hnRNP A1) is slightly stronger than that of wild-type TDP-43. Our findings provide insights into the structure of wild-type and ALS-associated mutants of TDP-43 in a cell.

## 1. Introduction

Amyotrophic lateral sclerosis (ALS) is a neurodegenerative disorder characterized by motor neuron dysfunction and muscle atrophy and is thus classified as a motor neuron disease (MND). A common feature of ALS is the formation of inclusion bodies (IBs) containing proteins and often RNA in the cytoplasm and nucleus of motor neurons [1]. Various proteins such as superoxide dismutase (SOD1), transactive response element DNA/RNA-binding protein 43 kDa (TDP-43), and fused in sarcoma (FUS) have been identified as components of IBs [2,3].

TDP-43 is the major disease protein in ALS as well as frontotemporal dementia (FTD) [3,4]. Many ALS-associated missense mutations in TDP-43 (*TARDBP*) that cause amino acid substitutions have been identified [3,5,6]. These TDP43 mutants are crucially involved in the onset and severity of ALS [2,7]. TDP43 carries two RNA/DNA-recognition motifs (RRM 1 and 2) that recognize single-stranded uracil/thymine- and guanine-rich nucleotides, an N-terminal ubiquitin-like dimerization domain (NTD), and a C-terminal intrinsically disordered region (IDR), including a prion-like Q/N-rich domain, which regulates protein interactions with other RNPs via a prion-like domain (e.g., heteronuclear ribonucleoprotein A1; hnRNP A1) (Figure 1) [6,8]. TDP43 harbors a nuclear localization signal sequence (NLS) between the N-terminal ubiquitin-like domain and RRM1, as well as a nuclear export signal sequence (NES) in RRM2. Thus, TDP43 is a nuclear-cytoplasmic shuttling and functional protein, involved in mRNA splicing, microRNA processing, and the transport of mRNA to the cytoplasm [3,9,10,11]. The structural basis of TDP-43 reinforces how these functions are regulated. Various domain structures of TDP-43 (e.g., NTD, RRM 1 and 2, and a portion of IDR), amyloid structures of peptides, and IDRs of TDP-43 have been identified [4,6,12,13]; however, the whole structure of TDP-43 has not yet been determined despite available enzymatic and interaction assays using soluble recombinant TDP-43 [14,15]. Furthermore, the structural determination of the amyloidogenic core region peptides in TDP-43′s IDR, using nuclear magnetic resonance (NMR) analysis, suggests that the ALS-associated mutation can change the conformation and angle of the IDR peptide [16]. However, it is still unclear whether this conformational change in the IDR, caused by the disease-causing amino acid mutation, affects the end-to-end conformation of the N- and C-termini of TDP-43.

Förster resonance energy transfer (FRET) is a physical mechanism that describes the energy transfer from the excited state of a donor fluorophore to the unexcited state of an acceptor fluorophore without the emission of donor fluorescence. The FRET efficiency mainly depends on three conditions: (i) displacement between the donor and acceptor fluorophores (within ~10 nm); (ii) spectral overlap between the donor fluorescence spectra and acceptor excitation spectra; and (iii) transition dipole moments of the respective fluorophores [17]. FRET is often used to detect conformational changes in proteins whose structures have not necessarily been identified (e.g., G protein-coupled receptors [GPCRs]) and the signal transduction states that accompany their interactions in live cells or solutions [18]. Therefore, FRET could be a potential method to provide structural insight into the whole TDP-43 structure in live cells with high time resolution. To improve FRET efficiency, circular permutations (CPs) in fluorescent proteins, in which the original N- and C-termini of fluorescent proteins are linked and new N- and C-termini are created by cutting the appropriate amide bond, have been employed [19]. CPs can improve the dynamic range of genetically encoded FRET-based sensors because the on-off dynamic range of FRET increases as the orientation between the fluorophores is appropriately modified [20]. Orientational rearrangement in the cytoplasmic aggregates of the ALS-associated mutant of SOD1 during proteasome inhibition and its recovery is detected using FRET between the monomeric teal fluorescent protein (mTFP1) and the monomeric yellow fluorescent protein Venus [21]. The change in FRET efficiency of a fluorophore pair when circular permutation was introduced suggests that the relative angle between fluorophores may be altered without changing their distance.

Fluorescence correlation spectroscopy (FCS) has been widely used to determine the diffusion coefficient of fluorescent molecules, not only in solution but also in live cells with single-molecule sensitivity [22,23,24]. The FCS can detect fluorescent molecules of interest with high sensitivity and specificity, even when cellular proteins are mixed in the solution and environment. Therefore, diffusion analysis can be performed without purifying the target proteins from the cell lysate. Fluorescence cross-correlation spectroscopy (FCCS) is an advanced FCS method that uses two-color fluorescence and has been used for biomolecular interactions in solution and live cells with single-molecule sensitivity [23,25].

In this study, we describe the structural basis of TDP-43 using FRET and FCS in live cells. The ALS-associated mutants of TDP-43 have a different conformation than their wild-type. The typical distance between the N- and C-termini and the molecular shape of TDP-43 in live cells was assessed. Furthermore, the interaction between ALS-associated TDP-43 and hnRNP A1 may be stronger than that with wild-type TDP-43.

## 2. Results and Discussion

### 2.1. The End-To-End Distance of TDP-43 between N- and C-Termini Using FRET

To examine the structural basis of TDP-43, FRET was used, which was altered by the relative distance and orientation between the donor and acceptor fluorophores. A *Clavularia*-derived monomeric teal fluorescent protein (mTFP1) as a FRET donor was tagged with TDP-43 at the N-termini, and a monomeric yellow fluorescent protein (Venus) as a FRET acceptor was tagged at its C-termini (hereafter, it is called T43V) (Figure 2b). To change the relative orientation between the two fluorophores, Venus tagged to T43V at the C-terminus as an acceptor was substituted with circularly permutated Venus (cp173Venus and cp195Venus) (hereafter referred to as T43V173 and T43V195, respectively) (Figure 2a,b). The efficient nuclear localization of T43V transiently expressed in the murine neuroblastoma Neuro2a (N2a) cells was confirmed using confocal laser scanning microscopy (CLSM) (Figure 2c). T43V was mainly distributed in the nucleoplasm but not in the nucleolus and often in bright nuclear bodies, as previously reported [23,26,27]. The FRET efficiency (*E*_FRET_) between the N- and C-terminal fluorophores in the nucleoplasm was measured by the fluorescence recovery of the donor fluorophore after acceptor photobleaching. To minimize the influence of intermolecular FRET of condensed TDP-43 in the nuclear bodies, the regions excluding the nuclear bodies and nucleoplasm were measured. *E*_FRET_ between the independently diffused mTFP1 and Venus monomers as a negative control was not observed (Figure 2d, lane 1). Although the ALS-associated mutations in TDP-43 did not affect its apparent localization and diffusion state (Appendix A), two frequently used mutants (A315T and Q331K) were selected for this study. All the *E*_FRET_ of the wild-type and ALS-associated mutants (A315T and Q331K) of T43V, T43V173, and T43V195 were statistically positive (Figure 2d, lanes 2–10). Although the *E*_FRET_ of T43V and T43V195 did not change regardless of the mutations A315T and Q331K (Figure 2d, lanes 2–4 and 8–10), the *E*_FRET_ of T43V173 was dramatically different between the mutants and wild-type (Figure 2d, lanes 5–7). A comparison between the Q331K mutants showed that the *E*_FRET_ of T43V173 was higher than that of T43V and T43V195 (Figure 2d, lanes 4, 7, and 10). The *E*_FRET_ depends on both the distance and the relative conformation [17]; however, TDP-43′s *E*_FRET_ was changed by the circular permutation of the acceptor fluorescent protein, which changes the relative angle between the two fluorophores without affecting their distance. The reason T43V and T43V195 did not show these changes in their *E*_FRET_ is likely because the dipole moment of Venus and cp195Venus shows a similar direction, but cp173Venus shows the opposite direction (green arrow in Figure 2b), indicating that the arrangement of the dipole moment of Venus and cp195Venus would be insensitive to changes in the conformation of TDP-43. However, these *E*_FRET_ changes in T43V173 suggest that the relative angle between the N- and C-termini of TDP-43 may be different for the wild-type, A315T, and Q331K mutants. Consequently, the ALS-associated mutation in the IDR of TDP-43 alters the relative angle between the N- and C-termini and not the end-to-end distance between the N- and C-termini of TDP-43 (*d*_NC_).

### 2.2. Possible Conformation of ALS-Associated Mutants and Wild Type of TDP-43

The positive *E*_FRET_ obtained in this study ranged from approximately 1.5% to 7.8% (Figure 2d). To determine whether these *E*_FRET_ values are appropriate for the distance between the N- and C-termini of TDP-43, the donor–acceptor distances were estimated from the *E*_FRET_. The distance between fluorophores is often calculated based on the assumption that the dipole orientation factor (κ^2^) is 0.67 [28,29]. Using this value, the Förster distance between the mTFP1 and Venus fluorophores, that is, the distance at which the energy transfer efficiency is 50%, was determined as 5.9 nm (Equation (2) in Materials and Methods). Based on this Förster distance, the distance between the fluorophores of the wild-type T43V (*d*^T43V^) was calculated as 9.6 nm using its *E*_FRET_ (5.3%; Figure 2d; hereafter, we continue to use this value as the typical *E*_FRET_ of T43V) (Equation (4) in Materials and Methods). The assumption for the κ^2^ is not precise here because the *E*_FRET_ was changed when substituting the circularly permutated acceptor (Figure 2d); however, it is challenging to determine the realistic κ^2^ in live cells. Therefore, we attempted to narrow down the *d*^T43V^ from the κ^2^. Based on the assumption that the κ^2^ has a maximum value of 4, the *d*^T43V^ should be 12.9 nm. In contrast, if the κ^2^ is 0.01, which is close to the minimum value because the FRET is not entirely observed when the κ^2^ is zero, the *d*^T43V^ can be assumed to be 4.8 nm. Hence, the *d*^T43V^ can be presumed to be in the range of approximately 4.8–12.9 nm. Furthermore, to consider whether the estimated range of *d*^T43V^ from the FRET analysis is a realistic value, we took into account the perspective of the actual TDP-43 structures. The solution structures of the individual domains and regions of TDP-43 (for example, the N-terminal Ubiquitin-like domain, RRM1&2, and a part of the IDR) have been determined [16,30,31,32]; however, all the structures of TDP-43, in which their domains are linked, have not yet been identified; therefore, we validated the *d*_NC_ using the full-length TDP-43 structure predicted by AlphaFold2 as a reference, while the structure of the IDR is most likely not highly precise. In this structure, the *d*_NC_ value of TDP-43 was determined to be 6.8 nm. Since fluorescent proteins are tagged in T43V, their tagging effects on distance must be considered. From the crystal structures of mTFP1 and Venus [33,34], the distance between the fluorophore and C-terminus in mTFP1 was determined to be approximately 1.9 nm, and the distance between the fluorophore and N-terminus in Venus was determined to be approximately 2.4 nm. If we assume that the distances between these fluorophores and their termini are simply added to the *d*_NC_ of TDP-43, the distance between the fluorophores is estimated to be 11.1 nm; the actual distance between the fluorophores is likely to be shorter than this value because the N- and C-termini of fluorescent proteins are unlikely to be in the most distant (opposite) orientation. In contrast, if the distance between the fluorophores is assumed to be 6.8 nm (i.e., the same as that of the *d*_NC_ predicted by AlphaFold2), the κ^2^ is calculated as 0.08 when the *E*_FRET_ is 5.3%. This κ^2^ is so close to zero, which is probably not supposed to be the case because this occurs only when the orientation between the fluorophores is highly restricted by strong steric hindrance between the N- and C-termini, such as in membranous proteins with no dramatic structural changes. The E_FRET_ changed according to the relative dipole orientation of the fluorophores by substituting the circularly permutated acceptors (Figure 2d); moreover, the IDR is not likely a rigid and uniform conformation because of its structurally disordered and diverse characteristics [35,36]. Therefore, the realistic κ^2^ can be assumed to be greater than 0.08, and the actual *d*^T43V^ could be longer than 6.8 nm. Furthermore, assuming that the peptide of the IDR (263–414 amino acids) is a fully extended linear chain, its length would reach ~50 nm. This is much longer than the longest distance between these fluorophores estimated from the *E*_FRET_ when the κ^2^ is 4 (12.9 nm), suggesting that the IDR of TDP-43 is not a stretched structure in the cells. On the other hand, the AlphaFold2-predicted *d*_NC_ of the N-terminal domains of TDP-43, including NTD and RRM1&2 (1–262 amino acids; NTD262), was 7.2 nm. This was longer than the *d*_NC_ of an intact TDP-43, as predicted by AlphaFold2 (6.8 nm). However, if the IDR adopts a conformation that turns from the C-terminus of NTD262 and is close to the N-terminus, the *d*_NC_ of a full-length TDP-43 can be shorter than that of NTD262. Therefore, the *E*_FRET_ observed here is not an inconsequential value, and *d*^T43V^ could be estimated in the range of approximately 6.8–12.9 nm (i.e., approximately 10 nm). Furthermore, the distance between the fluorophores of the T43V mutants associated with ALS (A315T and Q331K) would not be significantly different from that of the wild-type because the *E*_FRET_ values were similar.

### 2.3. Molecular Shape Estimation of T43V Using Fluorescence Correlation Spectroscopy

Following this, to examine the sphericity of the monomers or low-molecular-weight complexes of T43V present in the cell lysate, the fast diffusion coefficient of T43V was determined using FCS. To minimize the effect of intermolecular interactions on the apparent molecular weight, T43V was solubilized in a lysis buffer containing a high salt concentration and detergent with high ionic strength (Buffer H; see Section 3). The diffusion coefficients and their proportions of T43V, in addition to the TDP-43 tagged with eGFP (43G) and GFP monomers (eGFP) as controls, were obtained via nonlinear curve fittings using a two-component three-dimensional diffusion model for the autocorrelation function from the FCS measurements. All of the fast diffusion components of T43V, 43G, and eGFP were more than 93%; thus, they were dominant (Table 1). The fast diffusion coefficient of T43V was lower than that of 43G, indicating that it may reflect an increase in the molecular weight and a change in the molecular shape of T43V as a result of the mTFP1 tag at the N-terminus. Using the theoretical molecular weight of eGFP obtained from the amino acid composition (27.0 kDa) as a reference, the estimated molecular weights of T43V and 43G were estimated to be 500 and 270 kDa, respectively (Table 1). Derived from the molecular weight of the amino acid composition, based on the spherical assumption with eGFP as the reference, using the Stokes–Einstein equation, the theoretical diffusion coefficients of T43V and 43G were 52 and 66 μm^2^/s, respectively (Table 1). The Perrin factor, a ratio of the theoretical diffusion coefficient based on the spherical assumption and the FCS-measured diffusion coefficient, was then calculated to be 1.7 for T43V and 1.6 for 43G, which were both greater than 1 (Table 1), suggesting that the apparent molecular shape of T43V and 43G may be an ellipsoid, not a sphere. Although the mTFP1 tag at the N-terminus of T43V is expected to increase its ellipsoidal degree compared to 43G, the increase in the Perrin factors from 43G to T43V was only slight. Therefore, this ellipsoid shape could not be caused by the tagging of the fluorescent proteins at the N-and/or C-terminus of TDP-43 because TDP-43 would have an ellipsoid shape itself. Furthermore, the estimated molecular weights of TDP-43 using FCS (500 and 270 kDa) are likely invalid, and the actual apparent molecular weight is likely to be much smaller because the ellipsoid enhances the effect of friction with the solvent compared to the sphere.

### 2.4. Diffusion Coefficient of T43V in Live Cells Using Fluorescence Correlation Spectroscopy

In cells, it is easy to assume that TDP-43 can interact with various proteins and other intracellular biomolecules through diffusion via Brownian motion and collision. Since the FRET efficiencies of T43V were measured in the nucleus of live cells (Figure 2), the diffusion state of T43V, including the increase in its apparent molecular weight as a result of biomolecular interactions in the live cells, was examined using FCS. The diffusion coefficients and proportions of the wild-type T43V and its mutants associated with ALS (A315T and Q331K), in addition to the Venus monomers as a reference, were obtained via nonlinear curve fittings using a two-component three-dimensional diffusion model for the autocorrelation function from the FCS measurements. The fast diffusion coefficients of T43V were slightly lower than those of the Venus monomers (Table 2), suggesting a decrease in the diffusivity due to an increase in the amino acid components. The decrease in the fast diffusion coefficient of T43V in the live cells compared to that of the Venus monomers as a control was not as dramatic as that in vitro. Hence, the fast diffuse T43V in live cells may behave like monomers. The proportion of fast diffusion components of the wild-type T43V and its ALS-associated mutants was dominant (63–66%; Table 2), suggesting that the population of T43V that behaves like monomers may be high. Nevertheless, a slow diffusion coefficient was observed in all the samples (0.3–0.7 μm^2^/s; Table 2). The proportion of slow diffusion components of T43V (wild type and mutants associated with ALS) was significantly higher than that of the Venus monomers (Table 2), suggesting that, although the Venus monomers cannot efficiently interact with cellular biomolecules, the slow diffuse T43V can be involved in the dramatic increase in the apparent molecular weight as a result of the intermolecular interactions of TDP-43 (e.g., complex formation with cellular proteins) and most likely partially includes its self-oligomerization. Consequently, a considerable proportion of T43V may diffuse into the nucleus as monomers and/or low-molecular-weight complexes with molecular weights similar to those of the T43V monomers.

### 2.5. ALS-Associated Mutants of TDP-43 Strongly Interact with hnRNP A1

To elucidate how the ALS-associated mutations in the IDR of TDP-43 affect intermolecular interactions, we determined the interaction between TDP-43 and hnRNP A1 [37]. Colocalization between 43G and mCherry-tagged hnRNP A1 (RA1), transiently expressed in the Neuro2a cells, was confirmed in the nucleus using CLSM (Figure 3a). The interaction between 43G and RA1 was determined in the lysate of the Neuro2a cells containing a physiological concentration of cations (150 mM NaCl) using FCCS to detect the interactions of fluorescently labeled two biomolecules in solution [23,25]. The relative cross-correlation amplitude (RCA), an index of interaction strength, of a mixture of GFP and mCherry monomers as a negative control, was zero, but that of tandemly dimerized GFP and mCherry as a positive control was significantly high (Figure 3b, lanes 1 and 2). The RCA between the ALS-associated mutants (A315T and Q331K) of 43G and RA1 was significantly higher than that of wild-type 43G (Figure 3b, lanes 3–5). Although the IDR of TDP-43 is important for forming complexes with hnRNP A1 [37], a positive RCA between the IDR of TDP-43 alone (274–414 amino acids) and RA1 was not obtained (Figure 3b, lane 6), suggesting that RRMs are required for their interaction. Following this, because RNA is involved in the interaction between TDP-43 and hnRNP A1 [37], we determined the RCA after RNase treatment in the cell lysate. The RCAs of the negative and positive controls were unaffected by RNase addition (Figure 3b, lanes 7 and 8). However, the interaction between 43G and RA1 was lost after RNase addition (Figure 3b, lanes 9–11). RCA was not observed between the IDR and RA1 even after RNase addition (Figure 3b, lane 12). These results suggested that RNA binding to TDP-43 and hnRNP A1 is crucial for these interactions. To demonstrate that the ALS-associated mutations in TDP-43 are not involved in its interaction with RA1, cells co-expressing 43G and RA1 were lysed in concentrated NaCl-containing lysis buffers (190, 200, 210, and 220 mM NaCl), and FCCS measurements were performed. The RCA values of the negative and positive controls did not change with increasing NaCl concentration (Figure 3c, orange and cyan lines). The RCA values between 43G and RA1 decreased at 190 mM NaCl (~0.2) compared to that at 150 mM NaCl (~0.5, Figure 3b, lanes 3–5). In the case of wild-type 43G, the RCA reached a minimum (~0.1) at NaCl concentrations higher than 200 mM, suggesting that electrostatic interactions may contribute, but hydrophobic interactions may also be attributed to these interactions. However, in the case of the ALS-associated mutants of TDP-43 (A315T and Q331K), the RCA was still positive at 200 mM NaCl, but gradually decreased and was close to the minimum from 210 to 220 mM NaCl (Figure 3c, black, magenta, and green lines). Therefore, hydrophobic interactions in the ALS-associated mutants (at least A315T and Q331K) of TDP-43 with hnRNP A1 may contribute to a greater extent than in wild-type TDP-43. Because threonine is a neutral amino acid but lysine is not, the increased interaction may be involved not only in electrostatic potential but also in various interacting states through their overall structural changes (i.e., the conformational change of the IDR). This also supports the notion that various interactions via the conformational change of the IDR are likely important in the interaction between TDP-43 and cellular proteins.

## 3. Materials and Methods

### 3.1. Plasmids 

The plasmids for wild-type TDP-43 tagged with monomeric enhanced GFP (eGFP) carrying the A206K mutation (pTDP-43WT-eGFP), eGFP monomers (pTKbX-meGFP), iRFP713-tagged Histon H2B (pBOS-H2B-iRFP), and pCAGGS were described previously [23]. The plasmids encoding Venus carrying the monomeric A206K mutation and its circular permutants (cp173Venus and cp195Venus) were provided by Atsushi Miyawaki and Takeharu Nagai, respectively, as described previously [21]. The DNA fragment encoding Venus was inserted into the pEGFP-N1 vector (pVenus-N1). Fragments of DNA encoding cp173Venus and cp195Venus were inserted into pcDNA3.1(+) (pcDNA-cp173Venus and pcDNA-cp195Venus, respectively). The plasmid for mTFP1 has been described previously [21]. The plasmids for the ALS-associated mutants of TDP-43 were created by PCR-based site-directed mutagenesis using the plasmid for wild-type 43G as the template. The plasmid for FRET was created as the DNA fragments of mTFP1, TDP-43 (wild-type and A315T/Q331K mutants), and circularly permutated Venus were joined (T43V, T43V173, and T43V195) and subcloned into pcDNA3.1(+) (Thermo Fisher Scientific, Waltham, MA, USA) (pcDNA-T43V, pcDNA-T43V173, and pcDNA-T43V195). The DNA fragment encoding heteronuclear ribonucleoprotein A1 (hnRNPA1) was amplified from pET9d-hnRNP-A1 (#23026, Addgene) using PCR and inserted into mCherry-C1 (pmCherry-hnRNPA1). The plasmid for eGFP-tagged amino acids 274–414 of TDP-43 (peGFP-C1-C274) has been described previously [38]. All the DNA sequences were analyzed using a DNA Genetic Analyzer (ABI3130, Thermo Fisher Scientific).

### 3.2. Cell Culture and Transfection

The murine neuroblastoma Neuro-2a (N2a) cells were obtained from the American Type Culture Collection (ATCC; Manassas, VA, USA) and maintained in DMEM (D5796, Sigma-Aldrich, St. Louis, MO, USA), which was supplemented with 10% FBS (12676029, Thermo Fisher Scientific), 100 U/mL penicillin G (Sigma-Aldrich), and 0.1 mg/mL streptomycin (Sigma-Aldrich) as previously described [23,39]. One day before transfection, 2.0 × 10^5^ N2a cells were transferred to a glass-bottom dish (3910-035, IWAKI-AGC Technoglass, Shizuoka, Japan) for live cell imaging and FCS in the live cells, or a 35 mm plastic dish (150318, Thermo Fisher Scientific) for cell lysis. Plasmid DNAs were transfected using Lipofectamine 2000 (Thermo Fisher Scientific), according to the manufacturer’s protocol. To express the fluorescent protein-tagged TDP-43, a plasmid mixture with 0.3 mg of the one coding fluorescent protein-tagged TDP-43 and 0.7 mg of pCAGGS as a mock vector was used for the transfection. When iRFP-tagged H2B was co-expressed, 0.7 mg of pBOS-H2B-iRFP was added instead of pCAGGS to the plasmid mixture. To express monomeric fluorescent proteins, a plasmid mixture with 0.1 mg of the plasmid coding monomeric fluorescent proteins, and 0.9 mg of pCAGGS was used. To express GFP-tagged TDP-43 (43G) and mCherry-tagged hnRNPA1 (RA1), a plasmid mixture with 0.75 mg of pTDP-43WT-eGFP, 0.15 mg of pmCherry-hnRNPA1, and 0.1 mg of pCAGGS was used. To express the eGFP-tagged 274–414 amino acids of TDP-43 and mCherry-tagged hnRNPA1, a plasmid mixture with 0.8 mg of peGFP-C1-C274, 0.15 mg of pmCherry-hnRNPA1, and 0.05 mg of pCAGGS was used. Following overnight incubation, the transfection medium was replaced with the same medium that was used just before the observation for live cell imaging and FCS measurements in the live cells.

### 3.3. Confocal Fluorescence Microscopy

Confocal images were acquired using a laser scanning microscope system LSM 510 META + ConfoCor 3 (Carl Zeiss) equipped with a C-Apochromat 40×/1.2NA W Korr. UV-VIS-IR water immersion objective. To observe the nuclear localization of both T43V and H2B-iRFP, mTFP1, Venus, and mCherry were sequentially excited at 458, 514, and 594 nm, respectively. The excitation and emission waves were separated using the main dichroic beam splitter HFT458/514 for mTFP1 and Venus, and HFT488/594 for iRFP. To spectrally separate the fluorescence of mTFP1 and Venus, a secondary dichroic mirror NFT515 was used to split the fluorescence emission at 515 nm. mTFP1 fluorescence was detected after passing through a 470–495 nm band-pass filter (BP470-495), Venus fluorescence was detected after passing through a 530–575 nm band-pass filter (BP530-570), and iRFP fluorescence was detected after passing through a 655 nm long pass filter (LP655). The pinhole size was set as 92 mm. Fluorescence was detected using an avalanche photodiode (APD) detector. To observe colocalization between 43G and RA1, eGFP and mCherry were sequentially excited at 488 and 594 nm, respectively. The excitation and emission lights were separated using the main dichroic beam splitter (HFT488/594). To spectrally separate the fluorescence of eGFP and mCherry, a secondary dichroic mirror NFT600 was used to split the fluorescence emission at 600 nm. eGFP fluorescence was detected after passing through a 505–540 nm band-pass filter (BP505-540), and mCherry fluorescence was detected after passing through a 655 nm long-pass filter (LP655). The pinhole size was set as 71 mm. Fluorescence was detected using APD detectors.

### 3.4. Acceptor Photobleaching FRET

Confocal images were acquired using a laser scanning microscope system LSM 510 META (Carl Zeiss) equipped with a C-Apochromat 40×/1.2NA W Korr. UV-VIS-IR water immersion objective. mTFP1 and Venus were sequentially excited at 458 and 514 nm, respectively. The excitation and emission lights were separated using a main dichroic beam splitter (HFT458/514). Fluorescence was detected using a spectrometric multichannel detector; a range of 462–483 nm was collected for mTFP1 and 516–537 nm for Venus. The pinhole size was set as 248 mm. Fluorescence images were acquired by repeatedly irradiating a 514 nm laser at the maximum power after acquiring a single frame of the images. The FRET efficiency was calculated using the following equation:(1)$$E_{FRET}=\left[1-\frac{I_{T}^{pre}}{{C\cdot I}_{T}^{post}}\right]$$

In Equation (1), *E*_FRET_ is the FRET efficiency; $I_{T}^{post}$ is the fluorescence intensity of mTFP1, defined as when it reached a plateau after the photobleaching of Venus; $I_{T}^{pre}$ is the fluorescence intensity of mTFP1 before the photobleaching of Venus; C is the correction coefficient of the fluorescence loss of mTFP1 during acceptor photobleaching and image acquisition; this was obtained from the actual measurement of 20 cells expressing both mTFP1 and Venus in the live N2a cells.

The Förster distance between mTFP1 and Venus (*R*_0_) was calculated using the following equation:(2)$${R_{0}}^{6}=\frac{9000\ln\;(10)}{128\pi^{5}N_{A}}\frac{\kappa^{2}Q_{D}}{n^{4}}J$$
where *R*_0_ is the distance at which the energy transfer efficiency is 50% [17], *Q*_D_ is the fluorescence quantum yield of the donor in the absence of an acceptor, $\kappa^{2}$ is the dipole orientation factor, *n* is the refractive index of water, *N*_A_ is the Avogadro constant, and *J* is the spectral overlap integral calculated using the following equation:(3)$$J=\int \overset{-}{f_{D}}\varepsilon_{A}\left(\lambda \right)\lambda^{4}d\lambda$$

In Equation (3), $\overset{-}{f_{D}}$is the donor emission spectrum normalized to an area of 1, and *ε*_A_ is the acceptor molar extinction coefficient obtained from the absorption spectrum of the mTFP1 solution.

The distance between the fluorophores of wild-type T43V (*d*^T43V^) was calculated using the following equation:(4)$$d^{T43V}=\sqrt[{6}]{{R_{0}}^{6}\left(\frac{1}{E_{FRET}}-1\right)}$$
where $E_{FRET}$ is not a percentage but a proportion.

### 3.5. Protein Structure Analysis

PyMol Ver. 2.5.0 (Schrödinger, Inc., New York, NY, USA) was used to display and determine the distance between specific atoms in the models. The structure coordination files of TDP-43 and the fluorescent proteins were obtained from the AlphaFold2 server (https://alphafold.ebi.ac.uk/entry/Q13148) and a protein data bank. The end-to-end distance (*d*_NC_) was the alpha carbon of the first methionine and the 414th methionine or 262nd proline in the predicted structure of TDP-43. The distance between the N- or C-termini and the fluorophore in the fluorescent proteins was between the fluorophores and the alpha carbon of the 6th glycine at the N-terminus and the 221st asparagine at the C-terminus of mTFP1 (PDF#2hqk) or the 1st methionine at the N-terminus and the 230th tyrosine at the C-terminus of Venus (PDB#1myw).

### 3.6. Fluorescence Correlation Spectroscopy

To measure the diffusion coefficient in the live cells and cell lysates, FCS measurements were performed. For live cell measurements, the N2a cells transiently expressing 43G, T43V, or eGFP monomers were measured at 37 °C with 5% CO_2_ gas. The FCS measurements were performed using an LSM510 META ConfoCor3 system (Carl Zeiss, Jena, Germany) equipped with a C-Apochromat 40×/1.2NA W Korr. UV-VIS-IR water immersion objective, multi-line argon ion laser, and APD detectors. GFP and Venus were excited using the 488 nm laser line of the argon ion laser. Fluorescence was detected after passing through a 505 nm long-pass filter (LP505) using an APD detector. The pinhole was set to 70 mm. Autocorrelation functions were calculated for different lag times (t) and plotted as a function of the lag time to derive the corresponding autocorrelation curves. The autocorrelation curves were further analyzed by fitting a theoretical function for the three-dimensional diffusion of one component with blinking due only to equilibrium between the dark and bright states (Equation (5)) using the ZEN (Carl Zeiss) software:(5)$$\begin{array}{cc} G\left(\tau \right)= & 1+\frac{1}{N}\left[\frac{T}{1-T}exp\left(-\frac{\tau }{\tau_{T}}\right)\right]F_{Fast}\left[\frac{1}{{\left(1+\frac{\tau }{\tau_{D,Fast}}\right)}^{-1}{\left(1+s^{-2}\frac{\tau }{\tau_{D,Fast}}\right)}^{-\frac{1}{2}}}\right]\\ & +F_{Slow}\left[\frac{1}{{\left(1+\frac{\tau }{\tau_{D,Slow}}\right)}^{-1}{\left(1+s^{-2}\frac{\tau }{\tau_{D,Slow}}\right)}^{-\frac{1}{2}}}\right] \end{array}$$In Equation (5), *G*(τ) is the cross-correlation function; $\tau_{D,Fast}$ and $\tau_{D,Slow}$ are the fast and slow diffusion times of the samples, respectively; *F*_Fast_ and *F*_Slow_ are the fractions of the fast and slow components, respectively (*F*_Fast_ + *F*_Slow_ = 1; 100%); *N* is the average number of fluorescent molecules in the effective detection volume; *s* is the structure parameter representing the ratio of the radial (*w*_0_) and axial (*z*_0_) 1/*e*^2^ radii of the effective volume; *T* is the exponential relaxation fraction; and $\tau_{T}$ is the relaxation time of the fluorescence blinking dynamics. In the calibration experiments, the diffusion time and structure parameter (*s*) were determined using 0.1 mM ATTO488 (A488) as a standard. The diffusion coefficient of eGFP was calculated using the following equation:(6)$$D=\frac{\tau_{D}}{\tau_{A488}}D_{A488}$$In Equation (6), *D* is the diffusion coefficient of the samples, *D*_A488_ is the diffusion coefficient of A488 (400 μm^2^/s), and $\tau_{D}$ and $\tau_{A488}$ are the diffusion times of the samples and A488, respectively. The molecular weights of 43G and T43V were calculated using the following equation:(7)$$M_{FCS}={\left(\frac{\tau_{D}^{43}}{\tau_{D}^{GFP}}\right)}^{3}M_{w}^{GFP}$$

In Equation (7), $M_{FCS}$ is the molecular weight of the TDP-43 samples, and $M_{w}^{GFP}$ is the molecular weight of the GFP monomers (27.0 kDa); $\tau_{D}^{43}$ and $\tau_{D}^{GFP}$ are the diffusion times of the TDP-43 samples and GFP monomers, respectively.

The theoretical diffusion coefficient (*D*_Theo_), using the estimated molecular weight from the amino acid composition, was calculated using the following equation:(8)$$D_{Theo}={\left(\frac{M_{w}^{GFP}}{M_{w}^{AA}}\right)}^{3}D_{Fast}^{GFP}$$

In Equation (8), $M_{w}^{AA}$ is the theoretical molecular weight calculated from the amino acid composition of 43G and T43V, and $D_{Fast}^{GFP}$ is the diffusion coefficient of the eGFP monomers measured using FCS (Table 1).

To obtain the cell lysates, N2a cells transiently expressing Venus, wild-type T43V, and A315T/Q331K mutants of T43V were washed in PBS and thereafter lysed in a buffer containing 25 mM HEPES-KOH (pH 7.5), 500 mM NaCl, 1% NP-40, 1% sodium deoxycholate, 1% SDS, 0.01 U/mL benzonase, and 1% protease inhibitor cocktail (Sigma-Aldrich) (Buffer H). FCS measurements of Venus and Venus-tagged TDP-43 in the cell lysates were performed using an LSM510 META ConfoCor2 system (Carl Zeiss, Jena, Germany) equipped with a C-Apochromat 40×/1.2NA W Korr. UV-VIS-IR water immersion objective, multi-line argon ion laser, and APD detectors. Venus was excited using the 514 nm laser line of an argon ion laser. Fluorescence was detected after passing through a 530–610 nm band-pass filter (BP530-610) using an APD detector. The pinhole was set to 70 mm. The autocorrelation functions were analyzed in the same manner as those of live cells.

### 3.7. Fluorescence Cross-Correlation Spectroscopy

To obtain the cell lysates, N2a cells transiently expressing fluorescent proteins were washed in PBS and lysed in a buffer containing 50 mM HEPES-KOH (pH 7.5), 150 mM NaCl, 0.1% NP-40, and 1% protease inhibitor cocktail (Sigma-Aldrich). The interaction in highly concentrated salt was measured after adding high concentrations of NaCl (190, 200, 210, and 220 mM). RNase A (50 μg/mL; Nacalai Tesque, Kyoto, Japan) was treated before the measurement at 25 °C. FCCS measurements of the lysates were performed using an LSM510 META ConfoCor3 system (Carl Zeiss, Jena, Germany) equipped with a C-Apochromat 40×/1.2NA W Korr. UV-VIS-IR water immersion objective, multi-line argon ion laser, and APD detectors. GFP and mCherry were excited using 488 and 594 nm lasers, respectively. Fluorescence was detected after passing through a 530–610 nm band-pass filter (BP530-610) using an APD detector. The autocorrelation functions were analyzed in the same manner as those of live cells.

The GFP and mCherry fluorescent signals were separated using an NFT600 dichroic mirror. The GFP fluorescence was detected after passing through a 505–540 nm band-pass filter (BP505-540) using an APD detector. The mCherry fluorescence was detected after passing through a 655 nm long-pass filter (LP655) using an APD detector. The pinhole was set to 70 mm. The auto- and cross-correlation curves were further analyzed by fitting a theoretical function (Equation (5)) using the ZEN (Carl Zeiss) software. The RCA was calculated according to a previously reported procedure [23].

### 3.8. Statistics

The Student’s *t*-tests were calculated using Microsoft Excel.

## 4. Conclusions

In this study, we estimated the molecular shape of TDP-43 in live cells by using diffusion states and FRET measurements. The FRET analysis showed that the end-to-end distance of wild-type TDP-43 was approximately 10 nm. This estimation was not very different in the ALS-associated mutants (A315T and Q331K); however, the relative orientation of the dipole moment between the fluorophores suggests that the A315T and Q331K mutants of TDP-43 may have different conformations in live cells. Consequently, the different conformations of the ALS-associated mutants of TDP-43 may be due to a conformational change in the IDR-carrying mutation. Furthermore, the difference in IDR-mediated interactions of TDP-43 with cellular proteins, such as hnRNP A1, may be involved in the N- and C-terminal conformational changes in the ALS-associated mutants of TDP-43.

## Figures and Tables

**Figure 1 ijms-24-05513-f001:**
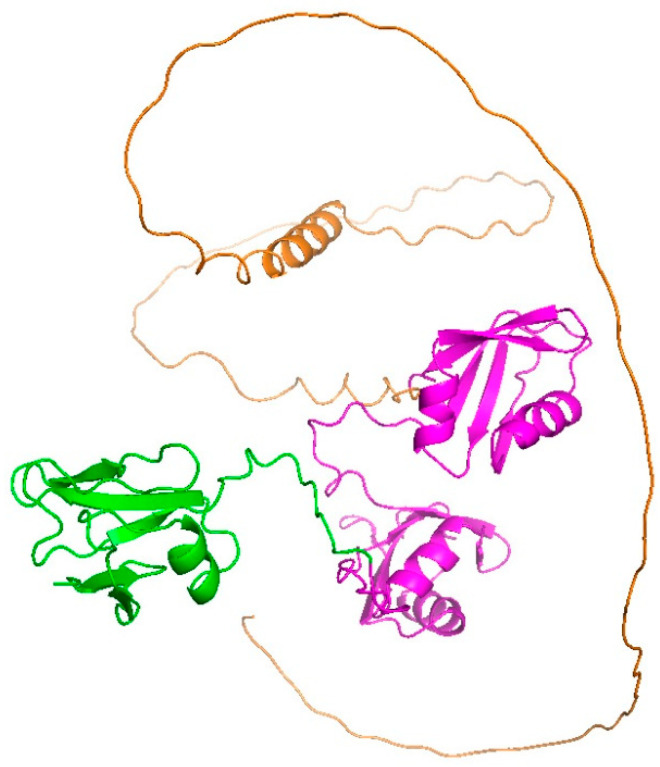
Domain structure of TDP-43 predicted by AlphaFold2 (Q13148). An N-terminal ubiquitin-like dimerization domain (NTD) (green), two RNA/DNA-recognition motifs (RRM 1 and 2) (magenta), and a C-terminal intrinsically disordered region (IDR), including a prion-like Q/N-rich domain (orange).

**Figure 2 ijms-24-05513-f002:**
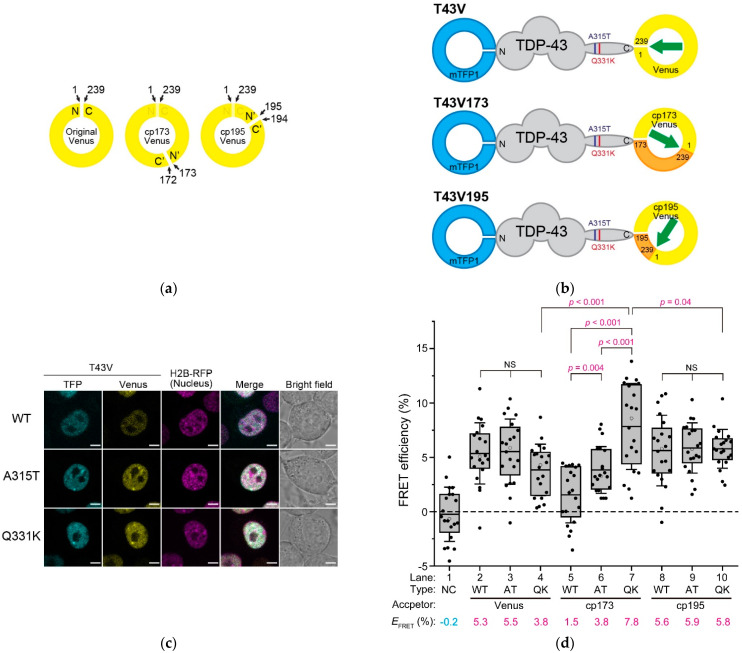
FRET efficiency of fluorescently labeled TDP-43. (**a**) schematic diagram of the circular permutation of an *Aequorea victoria*-derived fluorescent protein such as Venus. N and C denote the N- and C-termini of the original Venus. N’ and C’ denote the newly emerged N- and C-termini of the circularly permutated Venus (cp173Venus and cp195Venus). The numerical values and arrows indicate the positions of the amino acids; (**b**) schematic primary structures of the *Clavularia*-derived monomeric teal fluorescent protein (mTFP1)- and conventional or circularly permutated Venus-tagged TDP-43 (T43V, T43V173, and T43V195). V173 and V195 indicate the circularly permutated Venus (cp173Venus and cp195Venus, respectively). N and C denote the N- and C-termini of TDP-43. The positions of the ALS-associated mutations (A315T and Q331K) are denoted by purple and deep red color lines in the TDP-43 structure, respectively. N/C and the numbers in the illustration of Venus indicate the same as in (**a**). The green arrows indicate the schematic dipole moment of Venus; (**c**) confocal fluorescence images of the Neuro2A cells expressing T43V and Histon H2B-RFP as a nuclear marker. WT, A315T, and Q331K denote the wild-type and ALS-associated mutants of TDP-43, respectively. Scale bar = 5 μm; (**d**) box and dot plot of the acceptor photobleaching-based quantitative FRET efficiency of T43V in the live Neuro2a cells. The dots indicate the quantitative FRET efficiency measured in individual cells. The top and bottom of the box indicate the 75 and 25 percentiles, respectively. The horizontal lines in the boxes show the mean FRET efficiency of T43V as also indicated at the bottom of the graph (*E*_FRET_) (The number of analyzed cells = 20). The bars attached to the boxes indicate ± SD. The white circles indicate the median. The dotted line shows a zero value of the FRET efficiency. Type denotes wild-type TDP-43 (WT) or ALS-associated mutants, A315T (AT) and Q331K (QK), in addition to the co-expression of both mTFP1 and Venus monomers as a negative control (NC). Acceptor denotes a type of circular permutation of the yellow fluorescent protein as FRET acceptors: Venus, cp173Venus (cp173), or cp195Venus (cp195). Significance based on the Student’s *t*-test: NS (*p* ≥ 0.05); magenta color (*p* < 0.05).

**Figure 3 ijms-24-05513-f003:**
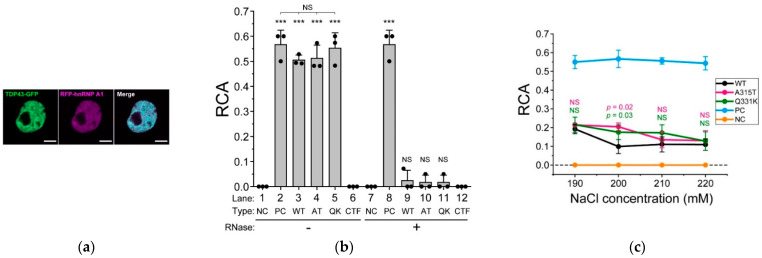
Biochemical interaction between TDP-43 and hnRNP A1. (**a**) confocal fluorescence images of Neuro2A cells expressing TDP-43-GFP and RFP-hnRNP A1. Scale bar = 5 μm; (**b**) relative cross-correlation amplitude (RCA) between TDP-43-GFP and RFP-hnRNP A1 measured using FCCS. The dots denote the independent three trials of a series of transfection, cell lysis, and FCCS measurement. The bars indicate the mean ± SD (three trials). Type denotes wild-type TDP-43 (WT) or its ALS-associated mutants; A315T (AT) and Q331K (QK), in addition to the co-expression of both GFP and RFP monomers as an interaction-negative control (NC), expression of tandemly dimerized GFP and RFP as an interaction-positive control (PC), and 274–414 amino acids carboxy-terminal fragment of TDP-43 (CTF). *** The Student’s *t*-test compared to the negative control (*p* < 0.001). NS: not significant (*p* ≥ 0.05); (**c**) comparison of the RCA between TDP-43-GFP and RFP-hnRNP A1 as the NaCl concentration is increased in the cell lysate (190, 200, 210, and 220 mM). RFP-hnRNP A1 was co-expressed with the wild-type (WT, black), A315T (magenta), or Q331K (green) mutants of TDP-43. A mixture of GFP and RFP monomers and tandemly dimerized GFP and RFP were used as an interaction-negative control (NC, orange) and an interaction-positive control (PC, cyan), respectively. The dots denote the independent three trials of a series of transfection, cell lysis, and FCCS measurement. The bars indicate the mean ± SD (3 trials). *p*-values above the bars: Student’s *t*-test compared to the RCA of the WT at the same NaCl concentration (the colors correspond to the ALS-associated mutants); NS: not significant (*p* ≥ 0.05).

**Table 1 ijms-24-05513-t001:** Diffusion coefficient and calculated molecular weight of fluorescent protein-tagged wild-type TDP-43 in cell lysate using fluorescence correlation spectroscopy.

Protein	*D*_Fast_ (μm^2^/s; Mean ± SD)	Fast Components (%; Mean ± SD)	*M*_FCS_ (kDa)	*M*_Theo_ (kDa)	*D*_Theo_ (μm^2^/s)	Perrin Factor
eGFP	91 ± 0.2	97 ± 0.3	-	27.0 ^†^	-	-
43G	42 ± 0.3	97 ± 0.4	270 ^⁋^	72.6 ^†^	66 *	1.6
T43V	30 ± 0.6	93 ± 2.2	500 ^⁋^	100 ^†^	52 *	1.7

^⁋^ The molecular weight was calculated from the diffusion coefficient measured using FCS. ^†^ Theoretical molecular weight calculated from the amino acid composition. * Theoretical diffusion coefficient calculated from the theoretical molecular weight.

**Table 2 ijms-24-05513-t002:** Comparison of the diffusion coefficient and calculated molecular weight of fluorescent protein-tagged wild type (WT) and ALS-associated mutants (A315T and Q331K) of TDP-43 tagged with mTFP1 and Venus (T43V) for FRET measurement in Neuro2a cells.

Protein	Mutant	*D*_Fast_ (μm^2^/s; Mean ± SD)	Fast Components (%; Mean ± SD)	*D*_Slow_ (μm^2^/s; Mean ± SD)	Slow Components (%; Mean ± SD)
Venus	-	43 ± 0.5	96 ± 0.3	0.7 ± 0.07	3.8 ± 0.3
T43V	WT	34 ± 0.5	66 ± 0.5	0.4 ± 0.01	34 ± 0.5
T43V	A315T	36 ± 0.3	63 ± 0.3	0.3 ± 0.01	37 ± 0.3
T43V	Q331K	37 ± 0.7	65 ± 0.5	0.4 ± 0.01	35 ± 0.7

## Data Availability

Not applicable.

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
