# Peer review of "Intracellular Conformation of Amyotrophic Lateral Sclerosis-Causative TDP-43"

_ijms, 2023, doi:10.3390/ijms24065513_

Round 1

Reviewer 1 Report

The paper is very interesting and clearly presented. I enjoyed reading all the sections, except the part of the distance calculations using as a basis the Fret efficiencies. I think this part is based on too many untested assumptions, and it is very speculative. I would suggest moving it to the discussion section or cutting it significantly.

Author Response

The paper is very interesting and clearly presented. I enjoyed reading all the sections, except the part of the distance calculations using as a basis the Fret efficiencies. I think this part is based on too many untested assumptions, and it is very speculative. I would suggest moving it to the discussion section or cutting it significantly.

[Response] We thank this referee for evaluating our manuscript. The comment of this referee is right that our manuscript includes not many but some assumptions. However, the distance from FRET efficiency is usually based on some assumptions, and these assumptions are still speculative if the entire protein structure has not been precisely determined. Thus, we believe that it makes sense to obtain the conformational information obtained using FRET on proteins whose structure has not yet been determined. We would like to share that the result of FRET efficiency itself is not based on assumptions. The point of moving it to the discussion section or not is that we have carefully considered it since we were preparing the draft of this manuscript. This manuscript is not long, and we believe that it is easy to interpret the result of FRET efficiency by discussing them immediately after explaining the result. Furthermore, as the other reviewers have not pointed out either the invalidity of some assumptions or that the discussion section should be independent, we would like to keep the structure of the paper as it is. We greatly appreciate you taking the time to seriously consider our manuscript. We hope that you forgive our conclusions.

Reviewer 2 Report

The authors of the manuscript studied a 43 kDa, 414 a.a. multi-domain transactive response element DNA/RNA-binding protein (TDP-43) and its amyotrophic lateral sclerosis -associated mutants. The structure of TDP-43 is only partially determined. Therefore, the aim of the authors of the publication was to determine the distances between the N and C ends of the protein, and to discuss the possible conformation of the wild-type protein and its changes in the ALS-associated mutant. To do this, they used the advanced biophysical methods: confocal microscopy and FRET between attached fluorescent proteins (mTFP1 and Venus) and fluorescence correlation spectroscopy in live cells.

The manuscript is written in an understandable way, the authors discuss well the obtained the results. It will be suitable for publication in the journal after the following corrections will be made.

TPD-43 is once written with a dash once without a dash. This should be standardized.

When describing the protein (line 37 et seq.), a Figure with the motifs highlighted (e.g. using AlphaFold and Pymol) will be helpful.

Figures placed in the manuscript should not be in a separate chapter, but should be placed progressively to illustrate the issues discussed.

Line 102:
“…with TDP-43 101 at the N- and C-termini, respectively (T43V) (Figure 1b).” The sentence style needs to be improved.

Line 107:
 The abbreviation CLSM appears for the first time. This is probably confocal laser scanning microscopy but it should be explained.

Line 114-116:
“Although the ALS-associated mutations in TDP-43 did not affect its apparent localization  and diffusion state (Figure S1), two typical mutants (A315T and Q331K) were selected for this study.”
What does "typical" mean? Why were these two mutants chosen out of the 11 tested?

Line 212
“…the estimated  molecular weights of T43V and 43G were estimated to be 500 and 270 kDa” Why are these values so high?

Figure 1(d):
 Please comment on why in some cases (e.g., lane 2,3,5,8) FRET efficiency values are below or close to zero.

Could you explain the difference in units between the Equation (2) and the equation in the cited reference [17], where instead of 20.7 is the value 9000ln(10) ~20.7*103?

Author Response

The manuscript is written in an understandable way, the authors discuss well the obtained the results. It will be suitable for publication in the journal after the following corrections will be made.

[Response] First of all, we thank this referee for evaluating and carefully reading our manuscript. According to this referee’s suggestions, we have modified the manuscript as follows.

TPD-43 is once written with a dash once without a dash. This should be standardized.

[Response] Thank you for the careful reading of our manuscript by this referee. We standardized it to ‘TDP-43’.

When describing the protein (line 37 et seq.), a Figure with the motifs highlighted (e.g. using AlphaFold and Pymol) will be helpful.

[Response] According to this referee’s suggestion, the predicted structure and its domains of TDP-43 are represented in Figure 1.

Figures placed in the manuscript should not be in a separate chapter, but should be placed progressively to illustrate the issues discussed.

[Response] Your point is valid. However, the submission format for Int. J. Mol. Sci. includes the separated figure sections (Section 2.2); thus, we are following this way in the first submission. We would like to discuss this point with the proofreading editor and appropriately modify the figure position before the online publication if it is accepted. We hope you will forgive our conclusions.

Line 102: “…with TDP-43 101 at the N- and C-termini, respectively (T43V) (Figure 1b).” The sentence style needs to be improved.

[Response] According to this referee’s suggestion, we have modified the sentences as follows:

[Lines 99–103] A Clavularia-derived monomeric teal fluorescent protein (mTFP1) as a FRET donor was tagged with TDP-43 at the N-termini, and a monomeric yellow fluorescent protein (Venus) as a FRET acceptor was tagged at its C-termini (hereafter it called T43V).

Line 107: The abbreviation CLSM appears for the first time. This is probably confocal laser scanning microscopy but it should be explained.

[Response] Sorry for our carelessness. We have modified it (Line 108).

Line 114-116: “Although the ALS-associated mutations in TDP-43 did not affect its apparent localization and diffusion state (Figure S1), two typical mutants (A315T and Q331K) were selected for this study.” What does "typical" mean? Why were these two mutants chosen out of the 11 tested?

[Response] So many ALS-associated mutations in TDP-43 have been reported that there is no single answer as to which mutations are appropriate. However, murine strains expressing mutants A315T and Q331K have been established and used as model mice for pathological analysis of ALS (for example, https://www.alzforum.org/research-models/als). These mice show ALS-like phenotypes. They have also been used in the analysis using cultured cells [10.3390/cells11152398; 10.1371/journal.pone.0081170]. However, after reconsidering the reviewer's question, we realize that the word “typical” was not appropriate and then is changed to “frequently used” [Lines 117–118]. Furthermore, our FRET-based conformational analysis method may play a role in clarifying the differences among such mutants.

Line 212 “…the estimated molecular weights of T43V and 43G were estimated to be 500 and 270 kDa” Why are these values so high?

[Response] This is likely because the ellipsoid shape enhances the effect of friction with the solvent compared to the sphere, resulting in an apparently large molecular weight. We interpreted this as a lack of explanation, and thus we have added the sentences as follows:

[Lines 226–229] Furthermore, the estimated molecular weights of TDP-43 using FCS (500 and 270 kDa) are likely invalid, and the actual apparent molecular weight is likely to be much smaller because the ellipsoid enhances the effect of friction with the solvent compared to the sphere.

Figure 1(d): Please comment on why in some cases (e.g., lane 2,3,5,8) FRET efficiency values are below or close to zero.

[Response] This is probably due to the effects of fluorescence photobleaching of mTFP as the donor. After acceptor photobleaching, the fluorescence intensity of the donor is recovered to the extent that it was lost by FRET, but it could be lower than before acceptor photobleaching according to the degree of fluorescence photobleaching during the acceptor photobleaching and their image acquisition. In this study, the effect of fluorescence photobleaching was corrected by the average fluorescence intensity after acceptor photobleaching in the control experiment (see Equation 1), but some of them may show negative values due to cell-specific errors.

Could you explain the difference in units between the Equation (2) and the equation in the cited reference [17], where instead of 20.7 is the value 9000ln(10) ~20.7*103?

[Response] This is adjusted according to the unit of length. We realize that it is better to use the same formula as the original equation in the textbook written by Dr. Lakowicz; thus, we modified it accordingly [Line 463].

Reviewer 3 Report

The present research article by Kitamura et al. entitled “Intracellular conformation of amyotrophic lateral sclerosis- 2 causative TDP-43” demonstrates the structural insight of TDP-43 protein and alterations in its confirmation due to ALS associated mutations by using live cell biochemical analysis with FRET and FCS spectroscopy methods. Overall data is quite promising. There are few queries that need to be addressed as listed below:

Major queries/comments:

#1. To study the biophysical structure of TDP43 in live cells, authors used Neuro-2a cells. Is there any rationale behind using this cell line?   

#2. Since BRET is another highly sensitive and precise method for protein interaction study, why author did not use this method in this study.

#3. TDP43 is a dynamic protein and exhibits nuclear-cytoplasmic shuttling function, however, authors did not conduct any relevant experiment to justify the correlation between TDP43 structure and its cellular function. Authors need to show TDP43 structural behavior in undifferentiated vs differentiated N2a cells to support their hypothesis in context to ALS.

#4. Besides nuclear localization, TDP43 is also shown to localize in other cellular organelles such as mitochondria. Authors need to show whether there is any structural alterations/heterogeneity in TDP43 conformations based on its intracellular localization in various cellular compartments.

Author Response

Overall data is quite promising. There are few queries that need to be addressed as listed below:

[Response] First of all, we thank you for pointing out that our data are “quite promising”. According to the constructive suggestions of this referee, we have modified our manuscript. We would appreciate if you could reconsider our revised manuscript.

#1. To study the biophysical structure of TDP43 in live cells, authors used Neuro-2a cells. Is there any rationale behind using this cell line?  

[Response] We chose it because they show high transfection efficiency. The high efficiency of transfection achieves the proportion of cells that efficiently express exogenous TDP-43, and its level of expression is easy to control. Therefore, since our previous reports, we use Neuro2a [10.1101/2022.07.03.498631; 10.1007/s12192-018-0930-1; 10.1371/journal.pone.0187813; 10.1038/srep19230, etc.]. Because TDP-43 is abundantly and endogenously expressed in Neuro2a, we believe that this is not an inappropriate cell strain to express TDP-43 exogenously. There have been many reports using Neuro2a for the analysis of TDP-43 [10.3390/cells11152398; 10.1371/journal.pone.0081170].

#2. Since BRET is another highly sensitive and precise method for protein interaction study, why author did not use this method in this study.

[Response] As this reviewer suggested, BRET is also a highly sensitive method to detect protein interaction. However, BRET has basically a lower time resolution than FRET. To maintain the long-term activation of the luminescence, expensive substrates for luciferase must be continually added to the culture medium. The future goal of this study is to track the conformational changes of TDP-43 in real time with high time resolution in live cells, and this led to the question of whether we can distinguish the conformational changes during the transition from the nucleus to the cytoplasm and in mitochondria, as you pointed out in the comment #4. Therefore, we here use FRET instead of BRET. We attempted this task with such a possibility in mind, but the FRET efficiency of T43V is not high, and more sensitive detection methods such as fluorescence lifetime imaging microscopy (FLIM) may be required to perform such observations. This has not been completed in this paper, and here we only report the validation data showing that averaged conformations of TDP-43 can be detected. We hope to clear some issues and to take on challenges to demonstrate them in the future.

In response to this reviewer's suggestion, we realize that the feature of FRET with high time resolution should be included in the Introduction as follows:

[Line 69–70] FRET could be a potential method to provide structural insight into the whole TDP-43 structure in live cells with high time resolution.

#3. TDP43 is a dynamic protein and exhibits nuclear-cytoplasmic shuttling function, however, authors did not conduct any relevant experiment to justify the correlation between TDP43 structure and its cellular function. Authors need to show TDP43 structural behavior in undifferentiated vs differentiated N2a cells to support their hypothesis in context to ALS.

[Response] The fluorescence intensity in the cytoplasm was very low because TDP-43 is mainly localized in the nucleus. As it is challenging to determine the FRET efficiency in the cytoplasm quantitatively, it is difficult to compare the FRET efficiency between the nucleus and the cytoplasm in the present situation. We would like to evaluate this using FLIM described in comment #2 as a future consideration. We also believe that the relevance to physiological relevance is important; thus, we agree with the use of the differentiation system. However, as Neuro2a cells cannot be differentiated into motor neurons, any system using motor neurons differentiated from iPSCs or other resources would be more pathologically applicable. We are in the process of establishing such differentiation and primary culture systems so that we hope to comprehensively elucidate the pathological relevance of the conformational changes of TDP-43 in the future. We believe that the data and validations in this manuscript are the cornerstones of these demonstrations.

#4. Besides nuclear localization, TDP43 is also shown to localize in other cellular organelles such as mitochondria. Authors need to show whether there is any structural alterations/heterogeneity in TDP43 conformations based on its intracellular localization in various cellular compartments.

[Response] Indeed, TDP-43 is reported to be localized also in mitochondria, but its abundance was not observed during our observations; therefore, it is difficult to detect FRET efficiency, as explained in #3. It must be very interesting to observe any structural alterations/heterogeneity in TDP43 conformations based on its intracellular localization in various cellular compartments as proposed by this reviewer, but it is currently technically difficult. Different cell lines and conditions must be newly established. Therefore, we would like this to be an issue in the future.

Finally, we are glad to have various constructive comments from this reviewer. We are sorry that a portion of your suggestions cannot be completed right now experimentally, but we sincerely hope that our future papers will provide solutions to these points.

Round 2

Reviewer 3 Report

I appreciate the precise response by authors to the comments/queries that I raised.